# A Limited and Diverse Set of Suppressor Mutations Restore Function to INX-8 Mutant Hemichannels in the *Caenorhabditis elegans* Somatic Gonad

**DOI:** 10.3390/biom10121655

**Published:** 2020-12-10

**Authors:** Todd Starich, David Greenstein

**Affiliations:** Department Genetics, Cell Biology and Development, University of Minnesota, Minneapolis, MN 55455, USA

**Keywords:** gap junctions, innexins, soma–germline interactions

## Abstract

In *Caenorhabditis elegans*, gap junctions couple cells of the somatic gonad with the germline to support germ cell proliferation and gametogenesis. A strong loss-of-function mutation (T239I) affects the second extracellular loop (EL2) of the somatic INX-8 hemichannel subunit. These mutant hemichannels form non-functional gap junctions with germline-expressed innexins. We conducted a genetic screen for suppressor mutations that restore germ cell proliferation in the T239I mutant background and isolated seven intragenic mutations, located in diverse domains of INX-8 but not the EL domains. These second-site mutations compensate for the original channel defect to varying degrees, from nearly complete wild-type rescue, to partial rescue of germline proliferation. One suppressor mutation (E350K) supports the innexin cryo-EM structural model that the channel pore opening is surrounded by a cytoplasmic dome. Two suppressor mutations (S9L and I36N) may form leaky channels that support germline proliferation but cause the demise of somatic sheath cells. Phenotypic analyses of three of the suppressors reveal an equivalency in the rescue of germline proliferation and comparable delays in gametogenesis but a graded rescue of fertility. The mutations described here may be useful for elucidating the biochemical pathways that produce the active biomolecules transiting through soma–germline gap junctions.

## 1. Introduction

Gap junctions are nearly ubiquitous in multi-cellular animals. The molecular constituents of gap junctions differ in chordates (connexins) and non-chordates (innexins), but their properties and biological functions are remarkably similar (recently reviewed in [1]). Although it is still unclear why a different class of gap junction molecule emerged within vertebrates, the ubiquity of gap junctions themselves suggests that being coupled is essential for most multi-cellular forms of life, and the functions of gap junctions are many. The sizes of connexin and innexin gene families within species further attest to a diversity of function. Gap junction coupling and the composition of junctional channels between cells can dramatically change over developmental time, though the rationale driving these changes is mostly unknown. We are studying the role that gap junctions play in the somatic control of germline development in *C. elegans*. Two symmetric gonad arms extend anterior or posterior of a central uterus and vulva (Figure 1). Initially, germ cells proliferate in a single small pool which becomes partitioned by migration of somatic cells from each of the developing gonad arms late in the second larval stage; these somatic cells eventually divide and develop to form the gonadal sheath, spermatheca, and uterus [2]. A somatic distal tip cell (DTC) occupies the leading edge of each expanding gonad arm, and the DTC establishes a stem cell niche supporting germ cell proliferation by producing Delta-class ligands LAG-2 and APX-1, which activate GLP-1/Notch receptors on germline stem cells [3,4,5,6]. At adulthood, the soma of a gonad arm includes the DTC, five pairs of sheath cells and the spermatheca. The Sh5 pair of sheath cells is the most proximal (closest to the uterus) and is connected to a constriction of the distal portion of the spermatheca. A series of sheath cell contractions, coordinated with the dilation of the distal constriction of the spermatheca, results in the ovulation of a maturing oocyte into the spermatheca, where fertilization and the completion of meiosis occurs (Figure 1).

The soma and germline are coupled by two classes of gap junction channels throughout development [7]. The somatic DTC and sheath express *inx-8* and *inx-9*, a pair of recently duplicated and redundant innexins, which constitute an operon (a single innexin is found at this locus in *C. briggsae*); either gene alone rescues the double *inx-8(0) inx-9(0)* knockout. Germ cells fail to proliferate in *inx-8(0) inx-9(0*) mutants (~4 per gonad arm). In the germline, *inx-14*, *inx-21*, and *inx-22* encode innexins that assemble into two classes of heteromeric hemichannels, INX14/INX-21 and INX-14/INX-22. *inx-14(0)* and *inx-21(0)* null mutants resemble *inx-8(0) inx-9(0)* in failing to support germ cell proliferation. In contrast, the *inx-22(0)* null mutant is fertile and its germline appears unaffected; however, feminized *inx-22(0)* mutants lacking sperm fail to inhibit oocyte meiotic maturation and ovulation in the absence of the meiotic maturation signal (major sperm protein, MSP [8]). Therefore, normal germ cell proliferation is likely to require the association of INX-8 and INX-9 with germline hemichannels containing INX-14 and INX-21. Thus, two classes of germline hemichannels (i.e., INX-14 with either INX-21 or INX-22) assemble into gap junctions with INX-8 and INX-9 in the soma to mediate multiple soma–germline interactions required for fertility—there is currently no evidence suggesting a function for isolated hemichannels, independent of gap junctions, in this system.

The failure of germ cells to proliferate in the absence of soma–germline gap junctions obscures any role that gap junctions may play in later germline developmental events. To address this issue, the *lag-2* promoter was used to express INX-8 in the DTC, but not sheath cells, in *inx-8(0) inx-9(0)* mutants [7]. In these animals, germ cell proliferation is restored to ~1/2 wild-type levels, but few progeny (avg. ~2) and a limited number of dead embryos (~20) are produced compared to the wild type (brood size ~300). This phenotype represents the ground state for the complete absence of sheath–germline gap-junctional coupling, and closely resembles that observed when sheath cell precursors (but not DTC) are ablated from developing gonad arms [9].

To explore more specific roles that the soma plays in nurturing the germline, we focus on manipulating *inx-8* somatic hemichannels in an *inx-9(0)* null background. Under these conditions, all gonadal gap junction functions are dependent on INX-8 in the soma. Ultimately, we would like to elucidate the nature of the molecules that move through soma–germline gap junctions to mediate their multiple functions. Because these active biomolecules are ill defined, and it is difficult to tag and trace their movement through gap junctions in vivo, we are using a genetic approach to identify interactions between innexins and candidate molecular pathways. To do so requires a palette of innexin mutations that may have differential effects on molecular trafficking. Here we describe the isolation of a series of non-null *inx-8* mutations derived from a genetic suppressor screen based on restoration of germ cell proliferation in a strong loss-of-function mutant affecting the second extracellular loop (EL2) of INX-8 (T239I). Such a screen was expected to potentially yield intra- or extragenic mutations. Intragenic mutations might provide new non-null *inx-8* mutants. Extragenic mutations might include compensatory mutations in germline innexins, or bypass mutations in biochemical pathways that rely on functional INX-8 channels. This latter possibility may be less likely unless a single gap junction-associated pathway is responsible for germ cell proliferation. Toward these ends, we conducted a large-scale genetic screen for suppressor mutations and isolated seven mutations, which represent intragenic second-site mutations conferring distinct channel properties to INX-8. The cryo-EM structure of the *C. elegans* INX-6 hemichannel was recently determined to be an octamer [10], and it has sufficient amino acid identity with INX-8 to estimate the positions of suppressor mutations in the INX-8 tertiary structure. Characterization of the phenotypes associated with suppressor mutations provides insights into the contribution of innexin domains to channel function and reveals a novel requirement for soma–germline gap junctions in the timely progression of gametogenesis.

## 2. Materials and Methods

### 2.1. Strains and Genetics

Worms were grown on standard NGM agar plates. All brood counts reported were performed at 20 °C. Bristol N2 was used as the wild type, and *mIs11 IV* was used to balance *inx-8 inx-9* mutants. Strain DG3954 *inx-8(tn1513) inx-9(ok1502); tnEx195[sur-5::gfp; inx-8(+) inx-9(+)]; tnIs107[inx-8p::mCherry; str-1::gfp]* served as the foundation for the suppressor screen. To screen for suppressors, DG3954 was mutagenized with EMS following standard protocols (50 mM, 4 hrs). Mutagenized hermaphrodites were plated singly, grown at 20 °C, and their F2 progeny were screened on a fluorescence dissecting stereomicroscope 5–6 days later for the presence of *sur-5::gfp(–)* animals with extended and reflexed *inx-8p::mCherry* gonad arms. Plates identified with such animals were then propagated to identify single hermaphrodites giving rise to candidate suppressors of *inx-8(tn1513) inx-9(0*). In total, 34 separate mutageneses were performed and 37,300 mutagenized hermaphrodites were plated. In seven mutageneses, a subset of at least 300 plates were scored for sterility, and the overall percentage of sterile or near-sterile mutagenized animals averaged ~35% (range 15–50%). If we assign a very conservative brood size of just 5 F1 progeny produced on average, this screen would represent ~240,00 mutagenized haploid genomes. These suppressors therefore represent rare mutations. Allele designations for suppressor mutations and corresponding amino acid changes follow (*tn1789* was used as the representative for E350K in brood counts as this mutation was independently isolated twice).

*inx-8(tn1513 tn1553) inx-9(ok1502)*– INX-8(T239I, M117T)*inx-8(tn1513 tn1555) inx-9(ok1502)*– INX-8(T239I, D24N)*inx-8(tn1513 tn1771) inx-9(ok1502)*– INX-8(T239I, S9L)*inx-8(tn1513 tn1789) inx-9(ok1502)*– INX-8(T239I, E350K)*inx-8(tn1513 tn1790) inx-9(ok1502)*– INX-8(T239I, A288V)*inx-8(tn1513 tn1791) inx-9(ok1502)*– INX-8(T239I, E350K)*inx-8(tn1513 tn1792) inx-9(ok1502)*– INX-8(T239I, I36N)

Delayed ovulation for INX-8(T239I, A288V) was determined as previously described [11]. Photos of phenotypes primarily employed a Zeiss motorized Axioplan 2 microscope with a 63x PlanApo (numerical aperture 1.4) objective lens and an AxioCam MRm camera (Carl Zeiss, White Plains, NY, USA) with AxioVision acquisition software (version. 4.8.2, Carl Zeiss, White Plains, NY, USA). Germ cell counts were made by photographing dissected, fixed, and DAPI-stained gonad arms. Sperm nuclei were not included in the final counts (highest counts were ~60 sperm nuclei).

### 2.2. Site-Directed Mutagenesis and Microinjection

*inx-8* mutations to be assayed for rescue of *inx-8(tn1474null) inx-9(ok1502null)*—abbreviated *inx-8(0) inx-9(0)*—on extrachromosomal arrays were synthesized using *Pfu* Ultra (Agilent). In most cases, a *Pst*I/*Nhe*I restriction fragment encompassing most of the *inx-8* coding region was used as template, and after sequencing to verify the desired changes, this fragment was subcloned back into an *inx-8::gfp* plasmid previously shown to rescue *inx-8(0) inx-9(0)* [7]. Plasmids were injected into N2 at 1–2 ng/μL along with a *str-1::gfp* marker (80–100 ng/μL). After stable lines were obtained, extrachromosomal arrays were crossed into *inx-8(0) inx-9(0)/mIs11* heterozygotes, and *inx-8(0) inx-9(0)* homozygotes expressing the array were examined for rescue. All constructs except INX-8(D24N)::GFP rescued i*nx-8(0) inx-9(0)* to some degree, indicating that they possessed a capability of forming junctions. To verify that INX-8(D24N)::GFP could not form gap junctions, we constructed the strain *inx-8(0) inx-9(0)/mIs11; tnEx222[lag-2p::inx-8::gfp; myo3p::mCherry]; tnEx223[str-1::gfp; inx-8(tn1555)::gfp].* In *inx-8(0) inx-9(0)* homozygotes, *lag-2p::inx-8::gfp* rescues germline proliferation but is only expressed in the DTC and not sheath cells. Gonad arms could then be dissected and antibody-stained for GFP and INX-22 [7], and any GFP signal in the proximal arm could be attributed to *inx-8(tn1555)::gfp*. Both GFP and INX-22 expression was evident in proximal arms, but no gap junction-like puncta were detected, suggesting that INX-8(D24N)::GFP cannot localize INX-22 into junctions.

INX-8(D332R)::GFP was similarly constructed and expressed in a wild-type (N2) background. In dissected gonads, its expression was detected using anti-GFP monoclonal antibody 3E6, while endogenous wild-type INX-8 and INX-22 were distinguished using antibodies specific to their respective C-termini, as previously described [7].

## 3. Results

We previously isolated a strong loss-of-function *inx-8(tn1513lf*) allele in a null *inx-9(0)* background [7], producing on average ~21 germ cells per gonad arm (cf. >1000 per wild-type arm [9]). t*n1513* encodes a T239I change in the second extracellular loop (EL2). INX-8(T239I)::GFP was introduced on a multi-copy extrachromosomal array into *inx-8(0) inx-9(0); Ex[inx-8(DTC+, Sheath–)].* In this background, *lag-2p* drives INX-8::GFP expression in the DTC and rescues germ cell proliferation; expression of INX-8::GFP in the sheath (especially apparent in the proximal arm) can be attributed to INX-8(T239I)::GFP. Importantly, proximal arm expression was sufficient to support localization of germline INX-22 to gap junction plaques, indicating that INX-8(T239I) is capable of assembling into somatic hemichannels and forming (non-functional) gap junctions with germline hemichannels [7]. INX-8(T239I) appears to cause a partial blockage in the channel it forms with the germline (total blockage might be expected to completely prohibit formation of channels).

Because INX-8(T239I)::GFP localizes to compromised channels, it was an ideal candidate for a suppressor screen. To carry out the screen, a strain was constructed including the following features: (1) the *inx-8(tn1513lf) inx-9(0)* mutation to be suppressed; (2) a means of visualizing the gonad at the dissecting microscope level, for which we designed an mCherry construct driven by the *inx-8* promoter (*inx-8p::mCherry*) that was integrated into the genome as a multi-copy array; and (3) an extrachromosomal array carrying wild-type copies of the *inx-8 inx-9* genomic region to rescue *inx-8(tn1513lf) inx-9(0)*, along with a co-injection marker (*sur-5::gfp*, expressed in the nuclei of all somatic cells), to visualize the presence of the array (Figure 2). Extrachromosomal arrays are frequently lost meiotically (and mitotically), resulting in a mixture of rescued and mutant progeny.

Hermaphrodites were mutagenized with EMS, placed singly on growth plates, and their F2 progeny were screened for signs of suppression. This was performed by scanning for evidence of germ cell proliferation—larger gonad arms—in animals which had lost *sur-5::gfp* (Figure 2). Seven independent suppressor mutations were isolated (from >200,000 mutagenized genomes, see Materials and Methods), representing six unique mutations. All were eventually confirmed as *inx-8* intragenic changes by linkage analysis and DNA sequencing. (For simplicity, we will refer to the suppressor strains by their amino acid changes rather than their genetic allele designations, which are listed in Materials and Methods.) For each suppressor mutation (SUP), we carried out the following analyses: (1) the phenotype of the INX-8(T239I, SUP) suppressor strain was characterized; (2) a corresponding INX-8(T239I, SUP)::GFP construct was introduced as a multi-copy extrachromosomal array into an *inx-8(0) inx-9(0)* background to see if the original suppression phenotype could be recapitulated; and (3) INX-8(SUP)::GFP lacking T239I was introduced on extrachromosomal arrays into *inx-8(0) inx-9(0)* to assay for rescue or novel phenotypes. The GFP tag allowed for verification of expression in the event that mosaicism might be suspected.

The identity of amino acid changes associated with the suppressor mutations, and their predicted topological locations based on homology to INX-6, are indicated in Figure 3. E350K was isolated twice, suggesting the screen was beginning to approach saturation. Mutations were found in three of the four TM domains, the N-terminus and the C-terminus. Although we expected that the starting T239I change in EL2 might cause blockage of the channel in the region of hemichannel–hemichannel association, none of the suppressor mutations were located in the extracellular loops. Likewise, no compensatory mutation in a germline innexin was isolated. We speculate that the suppressor mutations may relieve channel blockage by effecting a change in the EL tertiary structure within the INX-8 molecule, may increase access of molecules through the hemichannel to the site of the T239I blockage, or may do both. Suppressors are organized based on shared phenotypes.

### 3.1. Suppression Approximating Recovery to Wild Type

M117T (TM2) is the strongest suppressor, and the corresponding INX-8(T239I, M117T) hemichannel functions at nearly wild-type levels (Table 1). The cryo-EM model for INX-6 predicts that the narrowest constrictions in the innexin channel are in the region lined by the N-terminus, and the region spanned by the extracellular loops (specifically a helix in EL1; [10]). Evidence from tryptophan scanning mutagenesis of the Drosophila innexin Shaking-B(Lethal) TM1 domain suggested that TM domains are less tightly packed in the innexin channel than in connexin channels [12]. If so, specific alterations in TM domains might open the channel through shifting the packing arrangement of the TM regions, or by influencing the position of the N-terminus lining the channel. Because T239I, M117T hemichannels approach wild-type functionality, it is possible that the T239I blockage itself has been relieved by a resultant change in tertiary structure as well. Multi-copy arrays of INX-8(T239I, M117T)::GFP rescued *inx-8(0) inx-9(0)* to high levels of fertility.

### 3.2. Suppression with Reduced Germline Proliferation, Delayed Gametogenesis, and Reduced Brood Size

These mutants share approximately equivalent restoration of germline proliferation, exhibit a marked delay in gametogenesis, but display a graded rescue of brood size numbers. D24N is located in the N-terminus near TM1. It lies close to a second aspartate, D21, that by homology to INX-6, is predicted to contribute to anchoring of the N-terminus to the cytoplasmic dome [10]. We recently reported a detailed characterization of INX-8(T239I, D24N). Germline and brood sizes are ~1/3 of the wild-type N2 (Figure 4, and [11]). Gametogenesis and ovulation are delayed ~18 hrs in relation to the last larval (L4-to-adult) molt. Though delayed, once embryo production ensues it continues over several days, similar to the wild type. The rescue of *inx-8(0) inx-9(0)* by INX-8(T239I, D24N) extrachromosomal arrays recapitulates these phenotypes, though brood sizes are smaller (Table 1). Because INX-8(T239I, D24N) behaves as a strong reduction-of-function allele of *inx-8* that produces moderate brood sizes, it was used to establish genetic interactions with conditional mutants in the fatty acid synthesis pathway; this led to further genetic experiments demonstrating requisite transfer of malonyl-CoA from the somatic sheath to the germline to support early and continued embryogenesis [11].

A288V lies in TM4, and INX-8(T239I, A288V) gonad size and germ cell proliferation is slightly higher than that of INX-8(T239I, D24N) (Table 1; Figure 4), with lower levels (~2/3) of fertility (Table 1). INX-8(T239I, A288V) animals show a marked delay in gametogenesis in relation to the fourth larval molt similar in length to that seen for INX-8(T239I, D24N) [11], although a small percentage of animals failed to ovulate within 24 h of the molt (Figure 5). Multi-copy extrachromosomal arrays of INX-8(T239I, A288V)::GFP rescued *inx-8(0) inx-9(0)* to low levels of fertility (Table 1). We conclude that, in general, expression levels from the extrachromosomal arrays in these experiments appear lower than the levels derived from chromosomal expression.

E350K is located in the C-terminus. INX-8(T239I, E350K) gonad size and germline proliferation are comparable to INX-8(T239I, A288V), but virtually no viable progeny are produced (Table 1). Likewise, INX-8(T239I, E350K)::GFP extrachromosomal arrays rescue germline proliferation but not production of embryos in *inx-8(0) inx-9(0)* (Table 1). Lack of progeny prevented assessing the relationship of onset of ovulation to the L4 molt.

Inter-subunit interactions in the cytoplasmic domains of the INX-6 hemichannel are proposed to support the formation of a dome-like structure that surrounds the hemichannel pore face. Possibly disruption of this cytoplasmic dome by the E350K mutation may enlarge the pore face and allow increased access to the channel, which may allow for an increased probability of molecular transfer across the T239I-induced site of channel restriction. Amino acid sequence alignment of the 25 *C. elegans* innexins reveals that the most conserved regions lie in transmembrane domains and in sequences surrounding the invariant cysteines in the extracellular loops. The sizes and sequences of the cytoplasmic regions are more divergent, making residue-to-residue comparisons less certain. However, a more highly conserved C-terminal region within which INX-8(E350) resides includes an invariant aspartate (INX-8 D332; Figure 6). We engineered an INX-8(D332R)::GFP to assess the importance of D332; in otherwise wild-type animals, INX-8(D332R)::GFP fails to localize to gap junction plaques that are visualized by antibody staining with INX-22 and INX-8 antibodies (Figure 6). Intriguingly, a putative casein kinase II recognition site (SQSD) present in the *C. briggsae* homolog of INX-8/9 in this region is maintained in *C. elegans* INX-8 but not INX-9, and this region could be a site for regulation. We hypothesize that this C-terminal region maintains a conserved function in establishing cytoplasmic tertiary structure that is sufficiently perturbed by INX-8(E350K) to allow increased molecular access across the channel pore.

### 3.3. Suppressors with Defects in the Sh5 Pair of Somatic Sheath Cells

I36N (TM1) suppression of T239I is distinct from the other TM domain suppressors. Germ cell numbers are restored to ~1/5 wild-type levels in INX-8(T239I, I36N) (Table 1). Sperm and an occasional oocyte develop, but animals are sterile, due to a somatic defect. Late in the L4 larval stage, at a time when sperm are usually visible, it appears that the cytoplasm of the Sh5 pair of cells fills with fluid, and nuclei swell (Figure 7). Other fluid-filled foci can appear more distally in the gonad arm with time. Surprisingly, INX-8(T239I, I36N)::GFP expression from multi-copy arrays can rescue *inx-8(0) inx-9(0)* to low levels of fertility (Table 1). Of 36 gonad arms (18 animals) examined, only 2 arms resembled the phenotype of INX-8(T239I, I36N) animals. Overexpression of deleterious mutations on extrachromosomal arrays may preclude their recovery; thus *Ex[INX-8(T239I, I36N)::GFP]* arrays that rescue *inx-8(0) inx-9(0)* may be selected for lower expression levels of the transgene.

S9L lies in the N-terminus, and INX-8(T239I, S9L) mutants share with INX-8(T239I, I36N) a very small germline, the production of sperm but few oocytes, and swelling of somatic sheath cells. However, this swelling appears to be less restricted to the Sh5 pair of cells at onset (Figure 7). INX-8(T239I, S9L)::GFP arrays do not rescue *inx-8(0) inx-9(0)* to fertility, but unlike the original suppressor isolate, occasional fertilized embryos can be produced. Again, we hypothesize that the recovery of these arrays may be dependent on lower expression levels of the transgene compared to chromosomal expression of INX-8(T239I, S9L).

Studies of tryptophan-substitution mutations in the TM1 domain of the Drosophila ShakingB(Lethal) innexin identified several sites which exhibited increased conductance in relation to wild-type ShakingB(Lethal) [12]. These mutants also displayed open-hemichannel activity when expressed in unpaired Xenopus oocytes, which compromised oocyte survival; however, when paired, the survival of oocytes expressing these constructs improved, likely due to a reduction in free hemichannels as they assembled into gap junctions. Indeed, that opening of hemichannels may actually drive gap junction formation was originally proposed and demonstrated in a study of connexins expressed in Xenopus oocytes [15]. We speculate that I36N and S9L suppression of T239I results in hemichannels that are leaky in the cell membrane when unpaired. Sh5 expresses INX-8 at very high levels, at least partly due to the fact that upon ovulation the oocyte endocytoses gap junctions formed with the soma, thus depleting Sh5 of a significant number of INX-8/9 subunits that must be replaced [7]. Sh5 may therefore be especially susceptible to a defect resulting in open hemichannels. Additionally, because oocytes have not developed and advanced into the proximal arm at the time of Sh5 swelling, it appears that when hemichannels initially arise in Sh5 there are no available pairing partners in the germline with which they might form gap junction channels (sperm do not form gap junctions with sheath cells). This may be exacerbated if there is a delay in gametogenesis as seen with other suppressors, though we have not yet attempted to document this delay for I36N or S9L. The more severe phenotype of INX-8(T239I, S9L) suggests that other somatic cells that express *inx-8* at levels lower than Sh5 are also susceptible to this gain of function in the corresponding hemichannels, and S9L may create a greater degree of “openness” than I36N.

### 3.4. Expression of Suppressor Mutations in the Absence of T239I

Our interpretation of the nature of the suppressor mutations isolated in this screen is that they altered the T239I hemichannel sufficiently to restore some level of function. This might be by shifting TM domains, altering the N-terminus, or disrupting cytoplasmic pore size. As such, it was of interest to see if these mutations by themselves may alter hemichannels sufficiently to confer unusual phenotypes due to the resultant channels formed. We expressed each of the *inx-8* single suppressor mutations on extrachromosomal arrays to rescue *inx-8(0) inx-9(0)* mutants (Table 1). Because expression from such arrays can be variable, we focused on potential qualitative rather than quantitative effects.

As might be predicted, INX-8(M117T)::GFP rescued *inx-8(0) inx-9(0)* mutants robustly. Tagged versions of A288V, E350K, and I36N also rescued, but to reduced fertility. Multiple independent lines for each of these mutations were generated. No unusual phenotypes were identified that were seen consistently across each line representing a particular mutation, but expression levels may factor into whether such phenotypes might be recovered.

INX-8(S9L)::GFP arrays were difficult to recover. One balanced line gave rise to homozygous *inx-8(0) inx-9(0*); *Ex[INX-8(S9L)::GFP]* progeny in which germ cells proliferated but sheath cells displayed the characteristic swelling seen in INX-8(T239I, S9L) animals. A second independent INX-8(S9L)::GFP array in a balanced *inx-8(0) inx-9(0/++* background gave rise to animals with defective gonad arms and eventually could not be maintained except in a wild-type background, suggesting a dominant-negative effect. These results are consistent with the interpretation that S9L may lead to leaky hemichannel behavior that may be dominant to wild type in heteromeric hemichannels.

INX-8(D24N)::GFP showed no rescue of germ cell proliferation and was distinctive in being the only mutant tagged construct that appeared unable to contribute to channel formation. This was determined in two ways: (1) when expressed in *inx-8(0) inx-9(0); Ex[inx-8(DTC+, Sheath–)]*, INX-8(D24N)::GFP was unable to localize INX-22 to gap junction puncta in the proximal arm (see above for determination of T239I ability to form junctions, and Materials and Methods); and (2) whereas wild-type INX-8 and INX-8::GFP are endocytosed by maturing oocytes and can be identified in early embryos [7], INX-8(D24N)::GFP when expressed in a wild-type background was not detected in embryos. Therefore, D24N neither appears to make hemichannels on its own nor associates at significant levels with wild-type INX-8 to contribute to hemichannel formation.

## 4. Discussion

Germ cells in the adult gonad progress in an assembly-line fashion, providing a snapshot of continual developmental progression at any single time point. In combination with a detailed understanding of its development from embryo through larval stages, the *C. elegans* gonad offers an excellent model for examining the gap junction relationships and requirements between cell types in a structure from its origin to its final functional form. Although the composition of hemichannels in soma and germline do not change, we expect that the specific requirements for molecules passing through gap junction channels changes with developmental progression from a mitotic state through the stages of meiosis and gametogenesis; additionally, components moving through DTC–germline junctions may differ from those moving through sheath–germline junctions. Identifying molecules that transit through gap junctions in vivo is difficult, but genetic tools including mosaic analysis and mutations that perturb channel function have been useful in identifying candidates for traversing innexin gap junctions [11,16]. The isolation of this new set of *inx-8* mutations has provided insights into mutational consequences on the function of innexin domains as well as on gap junction communication between soma and germline.

Only a limited set of suppressors of INX-8(T239I) were isolated, but the suppressors were widely distributed within INX-8. This may suggest that mutations throughout the innexin subunit can affect the channel, but the nature of our screen selected only those that still allowed germ cell proliferation. That three of the suppressor mutations reside in different transmembrane domains is consistent with the model that TM domains are more loosely stacked in innexins than connexins [12]. The D24N, A288V, and E350K mutations, in the context of T239I, behave as reduction-of-function *inx-8* alleles. They show similar degrees of support for germline proliferation but vary in their effectiveness of rescuing production of progeny. The fact that T239I, E350K produces no viable progeny suggests that their corresponding channels could be more restrictive for molecule(s) required for healthy oocyte/embryo production versus molecule(s) required for germ cell proliferation. We cannot exclude the possibility that similar factors pass through DTC–germline and sheath–germline channels, with higher levels being required in growing oocytes versus germ cells. However, for example, malonyl-CoA production is required in the sheath for viable embryos, and expression of the enzyme for malonyl-CoA production (acetyl-CoA carboxylase) is not detected in the DTC [11]; malonyl-CoA could therefore be a candidate for a molecule whose passage from sheath to germline is particularly restricted by INX-8(T239I, E350K) hemichannels that can sustain proliferation but not production of viable progeny.

This set of suppressors revealed a gap junction requirement for timely gametogenesis, the nature of which is only a guess at present. Both sperm and oocyte development are delayed, but once gametes are produced, they appear to function normally—only approximately 10% of INX-8(T239I, D24N) embryos are defective [11]. This delay in gametogenesis could represent the inability to accumulate a necessary factor, or remove an inhibitory factor. The extent of this delay—almost a full day, remarkable for a generation time of ~3 days at 20 °C—is sufficiently long that suppressor screens for the restoration of wild-type timing of egg-laying are feasible and may uncover bypass mutations in the responsible pathway.

S9L and I36N suppression of T239I seem best explained as hemichannels that have open activity when unpaired. Clearly these hemichannels can form junctions, as they restore germ cell proliferation to some degree. The absence of a deleterious effect on the distal gonad arm suggests that most of the somatic hemichannels there are paired with germline hemichannels, or the levels of somatic hemichannels are sufficiently low that open hemichannel activity has a minimal effect on the plasma membrane. Expression levels of INX-8—graded from low in the distal arm to very high in the most proximal region [7]—are consistent with this being the determining factor in whether or not somatic sheath cell membrane integrity becomes compromised. A contributing factor for the Sh5 pair of cells, however, may be the absence of gap junction pairing partners for open hemichannels when they arise. It is possible such an explanation might apply to other widely expressed gap junction mutants with tissue-specific defects or with defects restricted to only a subset of expressing cells.

Other than being of structure-function interest, do these proposed open-hemichannel mutants have any utility for use in genetic interaction studies? Possibly. Because INX-8(T239I, I36N) arrays have been successfully isolated, by our interpretation due to lower expression levels, it would be possible to integrate these arrays into a chromosome and establish a reduction-of-function *inx-8* allele. Because these mutants are viable but display smaller germlines, they may be candidates for genetic interaction inquiries with other mutants affecting germline proliferation.

## 5. Conclusions

A genetic screen for rescue of germ cell proliferation in a strong loss-of-function INX-8(T239I) mutant yielded a set of suppressor mutations that restored channel function to varying degrees. The phenotypes of suppressor mutations that affect the cytoplasmic dome or appear to have open hemichannel activity provide supporting in vivo evidence for previous electrophysiological or biophysical studies on innexins reported by others. These mutations highlight distinct requirements for soma–germline coupling in the distal and proximal gonad arms and have helped identify a gap junction requirement for timely ovulation and gametogenesis that nevertheless does not inhibit the production of healthy broods. INX-8 (T239I, D24N) has already proved useful for demonstrating that malonyl-CoA moves through sheath–germline gap junctions; together, these suppressor mutations may facilitate the identification of additional molecules that are delivered through soma–germline gap junctions and their roles in promoting germline development.

## Figures and Tables

**Figure 1 biomolecules-10-01655-f001:**
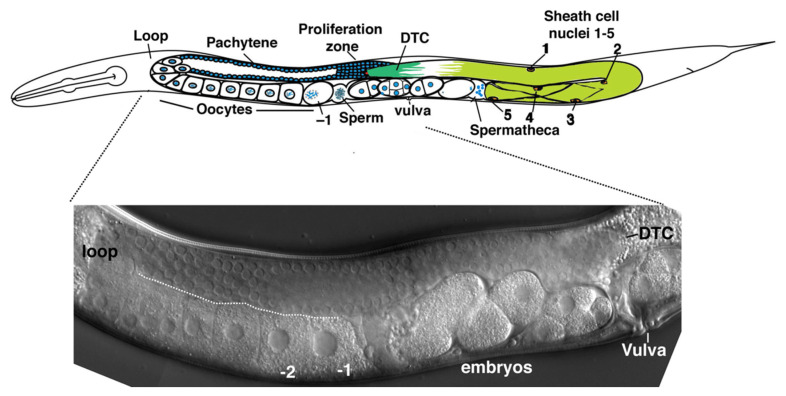
Diagram of the symmetric gonad arms in *C. elegans* (upper) and photo of corresponding gonad arm from the wild-type strain N2. DTC, distal tip cell; –1, most proximal oocyte.

**Figure 2 biomolecules-10-01655-f002:**
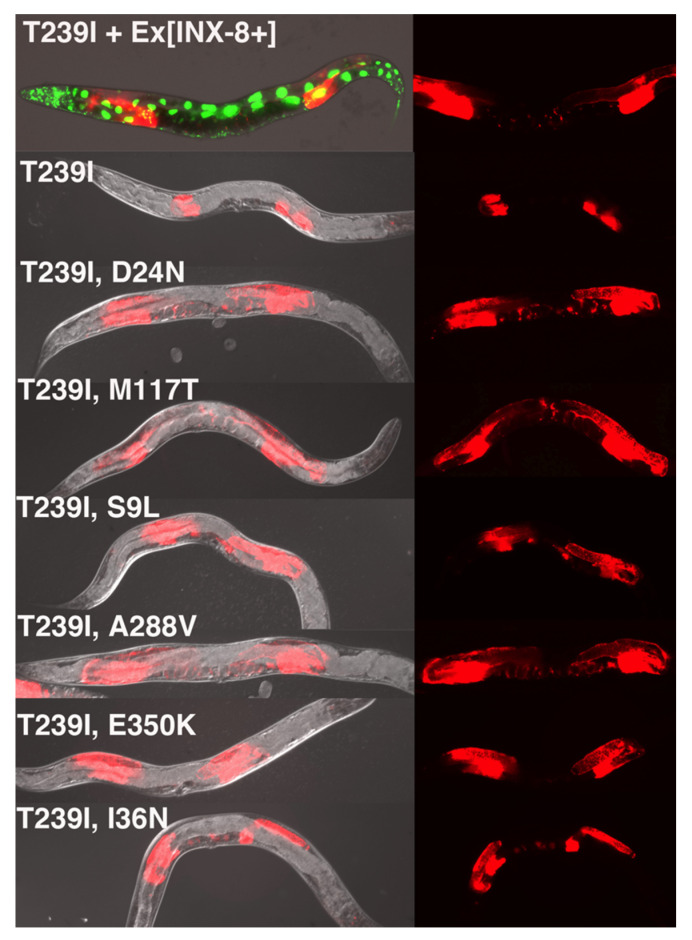
Suppressor mutants. The starting mutant, T239I, is rescued by an extrachromosomal array that carries wild-type *inx-8(+) inx-9(+)* and a *sur-5::gfp* co-injection marker (green). Meiotic loss of the extrachromosomal array is signified by a loss of GFP expression, which enables the degree of suppression to be assessed by the size of the gonad, as visualized using the *Pinx-8::mCherry* somatic gonad marker (red). Broods with F2 progeny exhibiting increased gonad size in relation to T239I were further examined. Adult worms are ~1 mm in length.

**Figure 3 biomolecules-10-01655-f003:**
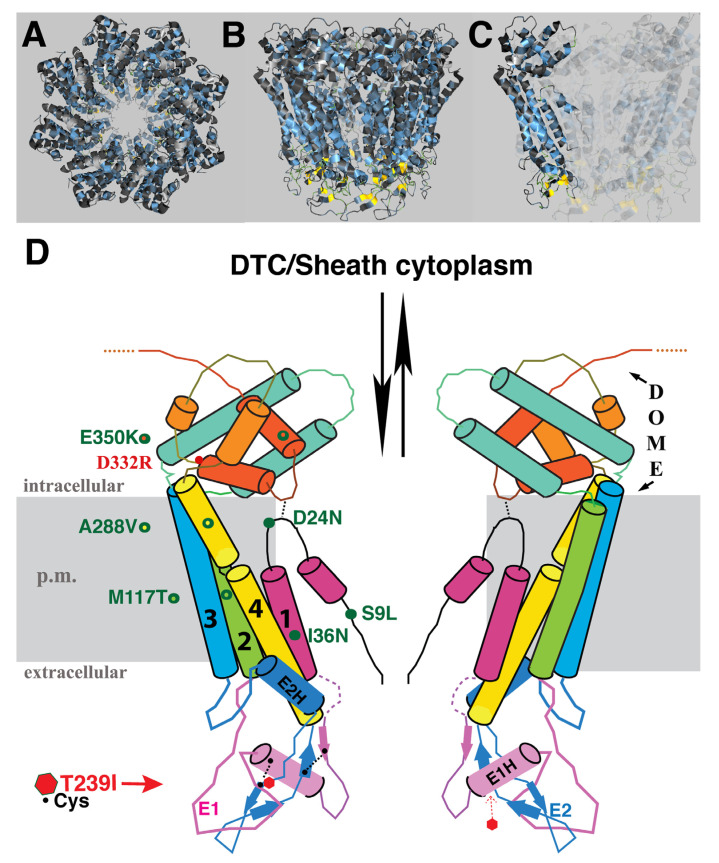
Positions of T239I suppressor mutations. (**A**–**C**) Proposed structural model of INX-8 hemichannels based on homology to INX-6 [10,13]. (**D**) Simplified representation of (**C**) to highlight predicted suppressor locations in relation to T239I. Position of the site-directed D332R mutation is also indicated. TM domains are labeled 1–4. E1, E2 = extracellular loops; p.m.= plasma membrane. Different colors are used to enhance contrast between domains for ease of visualization.

**Figure 4 biomolecules-10-01655-f004:**
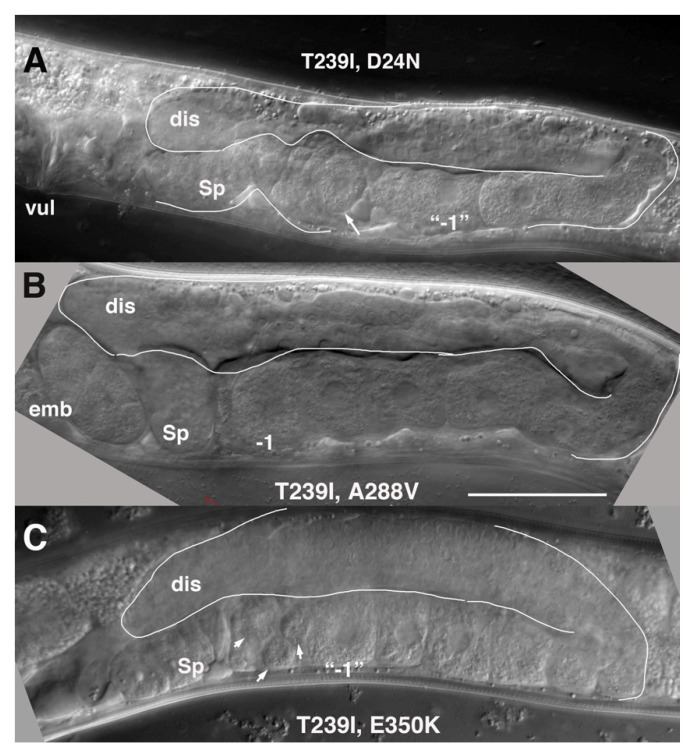
Representative gonad arms from young adults of suppressor mutants with reduced fertility and delayed ovulation in (**A**) INX-8(T239I, D24N), (**B**) INX-8(T239I, A288V), and (**C**) INX-8(T239I, E350K). The relatively smaller arms reflect reduced germ cell proliferation (compare to wild-type gonad arm as shown in Figure 1). Arrows in (**A**,**C**) point to unusually small oocytes in the most proximal position, a phenotype commonly seen in all three of these suppressors during early adulthood. Sp, spermatheca; dis, distal arm; vul, vulva; emb, embryos; −1, most proximal oocyte. Bar, 50 μm.

**Figure 5 biomolecules-10-01655-f005:**
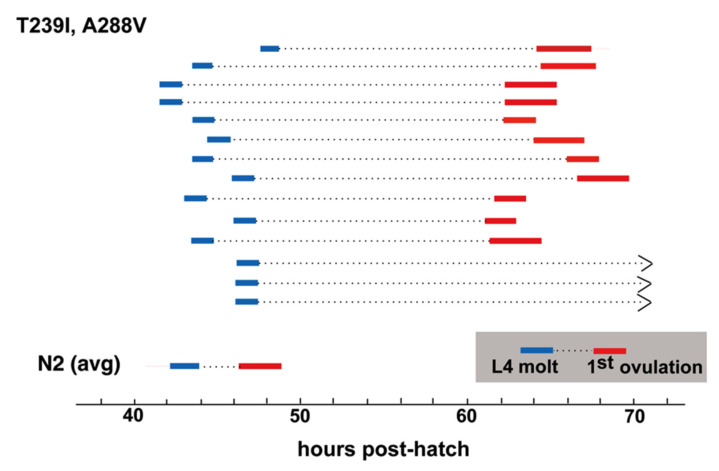
Ovulation is delayed in T239I, A288V mutants in relation to the L4 molt. The time between the last larval molt and first ovulation is ~3–5 h in the wild type. This period is greatly expanded in INX-8(T239I, A288V). (N2 avg. from [11].).

**Figure 6 biomolecules-10-01655-f006:**
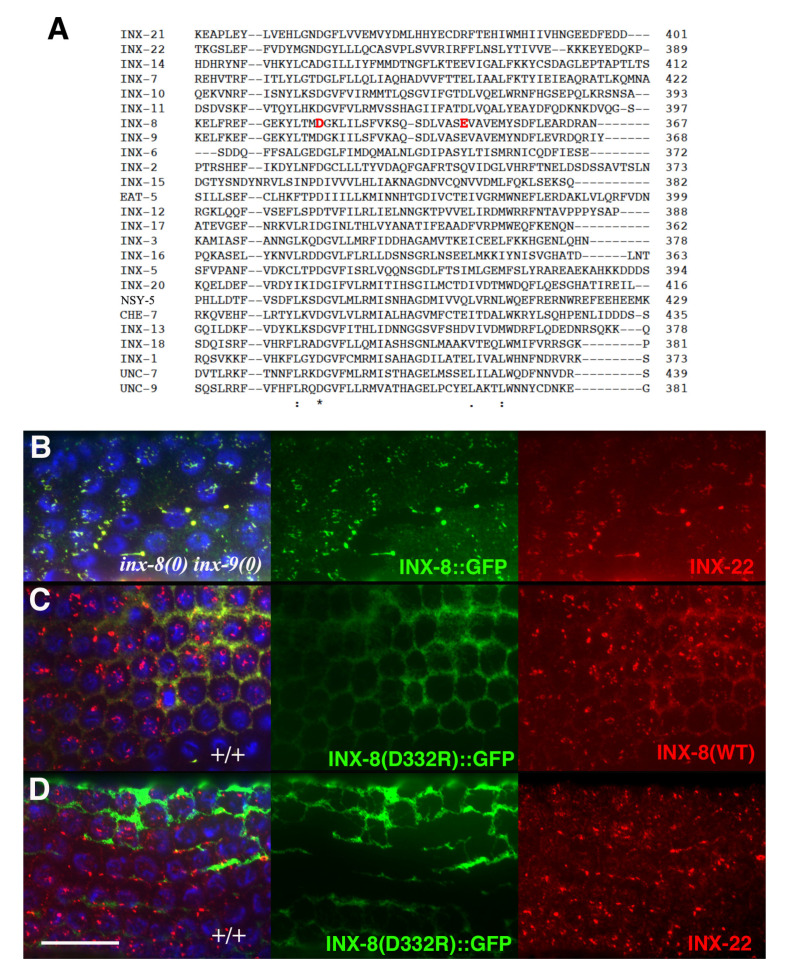
INX-8(D332R)::GFP fails to localize to gap junctions. (**A**) DNA sequence alignment of all *C. elegans* innexins (using Clustal Omega, [14]) reveals a conserved segment of the C-terminal region including E350 and an invariant aspartate D332 (highlighted in red). This aspartate is the only invariant residue in all of the cytoplasmic domains. (**B**) INX-8::GFP rescues *inx-8(0) inx-9(0)* and forms gap junction plaques (anti-GFP labeled puncta, green) co-stained with anti-INX-22 antibody (red) in the pachytene region of the distal gonad arm. (**C**) In a wild-type background, INX-8(D332R)::GFP in the pachytene region remains localized to the sheath membrane surrounding individual germ cells (honeycomb pattern, using anti-GFP, green). Antibody specific to the C-terminus of INX-8 detects wild-type INX-8 in puncta (red). INX-8(D332R)::GFP does not co-associate with wild-type INX-8 in plaques. (The weaker red honeycomb pattern may be due to some cross-reactivity with INX-8(D332R)::GFP, which still retains most of the C-terminus of INX-8 fused to GFP.) (**D**) In a wild-type background, INX-8(D332R)::GFP also fails to co-localize with INX-22 (red). Germ cell nuclei are stained with DAPI (blue). Scale bar, 20 μm.

**Figure 7 biomolecules-10-01655-f007:**
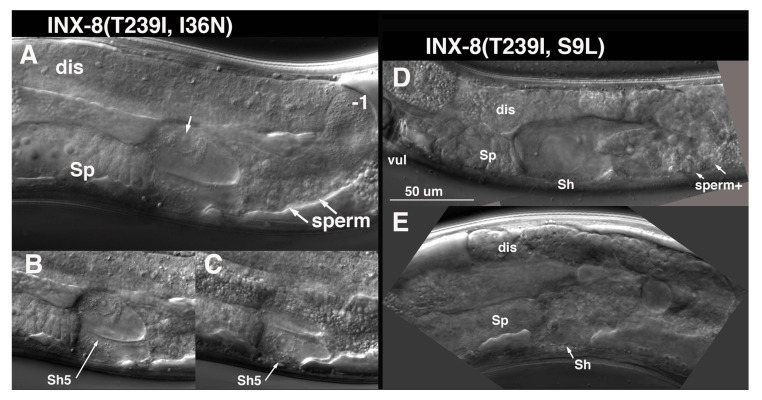
I36N and S9L affect cytoplasmic membrane integrity, especially of Sh5. (**A**–**C**) T239I, I36N mutants show swelling of both Sh5 cells (**B**,**C**), with nuclei visible within an expanded clear cytoplasm. Small central arrow in (A) indicates Sh5 nucleolus in enlarged nucleus with nuclear membrane still visible. Arrow in (B) points to plasma membrane of one Sh5 cell. (**D**,**E**) T239I, S9L hemichannels also affect Sh5, but loss of membrane integrity appears to include other cells or organelles as well. Sp, spermatheca; dis, distal arm; vul, vulva; Sh, sheath.

**Table 1 biomolecules-10-01655-t001:** Germline and Brood Sizes of INX-8(T239I) Suppressor Mutants.

INX-8 Amino Acid Changes	# Viable Progeny	Germ Cells/Arm
INX-8(T239I)	0	23 ± 20 (n = 76) ^1^
INX-8(T239, M117T)	256 ± 51 (n = 10)	ND
INX-8(T239I, D24N)	108 ± 40 (n = 90)	322 ± 18 (n = 3) ^2^
INX-8(T239I, A288V)	69 ± 20 (n = 15)	469 ± 31 (n = 3)
INX-8(T239I, E350K)	2 ± 2 (n = 15)	440 ± 85 (n = 3)
INX-8(T239I, S9L)	0 (n > 20)	173 ± 17 (n = 3)
INX-8(T239I, I36N)	0 (n > 20)	245 ± 38 (n = 3)

***inx-8(0) inx-9(0);* with extrachromosomal arrays containing:**		
INX-8(T239I, M117T)::GFP	ND	
INX-8(T239I, D24N)::GFP	48 ± 22 (n = 20)	
INX-8(T239I, A288V)::GFP	4 ± 4 (n = 30)	
INX-8(T239I, E350K)::GFP	0 (n > 15)	
INX-8(T239I, S9L)::GFP	0 (n > 15)	
INX-8(T239I, I36N)::GFP	62 ± 28 (n = 13)	

INX-8(M117T)::GFP	213 ± 62 (n = 10)	
INX-8(D24N)::GFP 2 lines	0 (n > 20)	
INX-8(A288V)::GFP line 1	78 ± 35 (n = 22)	
line 2	120 ± 48 (n = 21)	
INX-8(E350K)::GFP line 1	199 ± 47 (n = 9)	
line 2	101 ± 36 (n = 10)	
INX-8(I36N)::GFP line 1	110 ± 36 (n = 19)	
line 2	67 ± 43 (n = 10)	
INX-8(S9L)::GFP 2 lines	0 (n > 20)	

^1^ From [7]; ^2^ from [11]**.**

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
