# Peer review of "A Limited and Diverse Set of Suppressor Mutations Restore Function to INX-8 Mutant Hemichannels in the Caenorhabditis elegans Somatic Gonad"

_biomolecules, 2020, doi:10.3390/biom10121655_

Round 1
Reviewer 1 Report
Manuscript ID: biomolecules-1004599
Title: Mutations that affect channel opening of innexin hemichannels in the C. elegans gonad
Starich and Greenstein have reported the mutants that are associated with restoration of germ cell proliferation of loss-of-function mutant, T239I of INX-8. The authors have investigated suppressor mutations which are far from T239I located in EL2, and represented different degree of rescue from WT to partial of germline proliferation. Based on the phenotypes, they interpreted the effects for three mutants. E350K likely disrupts the cytoplasmic dome of INX-8, which leads to the increase of permeant flux through the pore. S9L and I36N would have open activity hemichannels, restoring the germ cell proliferation, and eliminate somatic sheath cells. These and other mutants are useful for understanding the relationships between gap junctions and germline development.
Major:
Introduction is too long and even more difficult to understand for those outside the field. What exactly the authors want to know from the experimental design taken in this study is unclear. The authors use T239I, a strong loss-of-function inx-8 mutant, as a baseline. What is the point of rescuing this mutation? The biological significance of restoring the function of this particular mutant, T239I, should be clearly stated. The introduction should be revised to be simpler, with a clearer description of the research purpose, so that everyone can understand it.
The provided data are numbers of viable progeny, proliferated germ cells, but there is no functional analysis of each mutant. Something like electrophysiology or dye transfer analysis if the functional properties of these mutant are discussed. In the absence of these, discussions in terms of channel functionality can only be very limited. The authors’ interpretation about E350K is that it causes disruption of the cytoplasmic dome and leads to increase flux through the pore, but unlikely to affect the channel constriction in T239I. Possibly E350 may have intersubunit interaction with the adjacent cytoplasmic domain. Even though, why could the loss of only this single hydrogen bond be the cause of the disruption of the cytoplasmic dome, which may enlarge the pore face? There are several other subunit interactions that have been formed, not only E350. Also, let’s say that E350K disrupts the cytoplasmic dome leading to enlarge the pore. What is the foundation that flux increases under such situation? It is too speculative to say that E350K is unlikely to affect the channel constriction of T239 mutant. When the cytoplasmic domain breaks, the constriction of the channel may also break, resulting in no functionality. This should be addressed.
The arguments sound like based on hemichannels, not gap junction channels of INX-8. Why can the authors say that phenotype recovery is due to only hemichannel activity? It is not clear how to distinguish between gap junction function and hemichannel function. Does INX-8 form gap junction or non-junctional hemichannel in vivo? The formation and functionality of hemichannel or gap junction channel should be considered separately. Moreover, if my understanding is correct, expression of INX-8 and INX-9 is overlapped in the gonad, and their function is complementary. In ref. 7, it is described that expression of INX-8, -9, -14, -21, and -22 overlapped closely throughout the gonad. Is there any reason that it is possible to ignore the other INXs? What if the phenotype rescue by INX8(T239L, S9L) is derived from INX-9 which is a component of heteromeric hemichannel with INX-8, not leaky function of INX-8 itself? These should be clearly addressed.
The figure legend is unfriendly. In particular, there is little explanation of what each of the sub-panels means and thus readers have to guess. It should be described in a way that the reader can understand what each color in fluorescence micrographs represents, and what (A), (B), and (C) mean, respectively.
Minor:
Fig.3 legend
…..Simplified representation of (C))à (C) (duplicated parenthesis)
Author Response
Response to Reviewer 1
Thank you for taking the time to review our manuscript. We very much appreciate the constructive comments on our manuscript and we have taken them to heart to improve the quality of the presentation as follows.
- The reviewer felt that our Introduction was difficult to understand for readers outside the field.
Response: We have reduced the usage of jargon, edited the text extensively for clarity, and streamlined the presentation in the Introduction. We think that readers outside the field will appreciate the juxtaposition of the genetic analysis with the structural interpretations of the mutants.
- The reviewer felt that the Introduction should emphasize the purpose of the research.
Response: We have rewritten the Introduction to emphasize the rationale for taking a genetic approach towards elucidating the nature of the molecules that transit through gap junctions in vivo to mediate their biological functions. The strength of this approach is highlighted by our recent publication in eLife (Starich et al., 2020), which used genetic tools described in the current submission to provide evidence that malonyl-CoA transits through soma-germline gap junctions to mediate a subset of their biological activities. We think it likely that many of the genetic tools described and developed here will prove similarly useful and we have modified the Discussion to make this point more clearly.
- The reviewer felt that the figure legends were not reader friendly.
Response: We have edited the figure legends for clarity and thank the reviewer for pointing out this issue.
- The reviewer was concerned about our usage of “hemichannels.”
Response: We have clarified that we use “hemichannels” to refer to the individual set of innexin octamers that contribute to the formation of gap junctions between the two coupled cells (e.g., in the wild type, INX-8/INX-9 hemichannels in the soma and alternatively INX-14/INX-21 or INX-14/INX-22 hemichannels in the germline). To avoid confusion, we now explicitly state, “there is currently no evidence suggesting a function for isolated hemichannels, independent of gap junctions, in this system.” That said, we provide novel evidence that certain mutant INX-8 hemichannels are toxic to somatic cells, likely because they are open and disrupt osmotic balance of the cells in situations during spermatogenesis when partner hemichannels in the germline are unavailable (see Figure 7).
- The reviewer questioned how a single change in charge, as exemplified by the E350K mutation, could appreciably affect the cytoplasmic dome.
Response: We have added a new figure (Figure 6), which demonstrates that changing a single conserved aspartic acid residue to an arginine (D332R) within the dome region has a drastic effect and renders the resultant hemichannels unable to form gap junction channels.
- The reviewer asked us to conduct dye injection or electrophysiological experiments to functionally assess the mutant channels described in our manuscript.
Response: We respectfully disagree that such experiments are necessary for understanding the in vivo roles and functions of gap junctions. Regarding dye injection studies, such experiments only address what molecules can move through gap junctions, not what molecules must move through gap junctions in vivo to mediate their biological functions. Further, in this system our unpublished data show that dye injection studies are problematic because the gonadal sheath cells are thin and easily damaged by injections, causing the dye to leak into the germline. Regarding electrophysiology, such studies report electrical coupling but do not address the transit of selective metabolites, such as malonyl-CoA, shown to mediate some of the biological effects of gap junctions in this system. We are unaware of any published reports in which electrophysiology was applied to intact C. elegans gonads in living animals. Because electrophysiological experiments have not been attempted in this system, we have been careful to steer clear of using electrophysiological terminology, such as “conductance” or “gating.” Instead, we have used “openness” in places in which we speculate on an intersection of biological function with the structural and electrophysiological studies of other innexins.
We differ with the reviewer on the general perspective of what constitutes “functional analysis.” We would argue that measurement of germ cell proliferation and brood sizes represent functional analysis at the biological level, as these measurements provide an indication of how well the mutant gap junctions function in vivo. Although it would be of great interest to know how electrophysiological properties or tertiary structure are altered in these mutant hemichannels, this would involve reconstructing the channels in an “artificial setting,” such as in Xenopus oocytes or as purified proteins without auxiliary elements. Although such studies could provide invaluable information, one might argue that determination of properties in these studies does not necessarily speak to in vivo function, especially in different tissue settings and with different interacting proteins.
Reviewer 2 Report
Interesting that the mutagenesis screen only delivered intragenic revertants affecting INX-8, and not any mutations in other innexin genes. Suggests strongly that the composition of the GJs between germline and sheath cells for this metabolic rescue are strictly limited in potential membership, and that no other innexin gene can jump into the breech to rescue the phenotype. Am I interpreting this correctly from your results? No rescues come from effects on tissue expression or sequences of cooperating innexin genes either from the sheath cell or from the germline side? I’m wondering particularly whether alterations of innexins from the germline side could help to “open up’ the channel with a defective INX-8 on the sheath side.
I do not see any mention of the status of the distal tip cell in these mutants, before or after rescue? Can’t you see it in your marked strains?
Table 1. Please explain why the germline in some rescued mutants can show so many germ cells per arm, yet produce so few viable progeny. Are there also defects in sperm viability? Can the “delayed germline” phenotypes be rescued via mating with wild type males to provide better sperm?
Figure 3 could be improved by indicating the cytoplasmic and apical sides of the membrane (maybe indicate intracellular vs extracellular), and the locale of the putative “dome”
In describing the phenotype of the I36N rescue strain, you mention that nuclei appear to swell at one point. Are these the sheath cell nuclei, or germline cell nuclei, or both?
When I view Figure 6, it seems that sperm are failing to move past sh5 towards the spermatheca? Is this a temporary delay, or does this remain true into adulthood? What are the ages of the animals in this Figure?
In the long run, it will be interesting to introduce targeted changes to the INX-9 channel in the same place as the T239I locale, to see if that would cause a similar germline phenotype in worms expressing only INX-9 in the sheath with INX-8(0).
Is there anything known about other membrane proteins that make up the rest of the dome?
Author Response
Response to Reviewer 2
Thank you for your insightful comments! It is helpful to have questions to see where we may include more explanatory material.
- Interesting that the mutagenesis screen only delivered intragenic revertants affecting INX-8, and not any mutations in other innexin genes.
Response: It is true that our suppressor screen might have conceivably picked up extragenic mutations, and because of the position of the T239I mutation in INX-8 it seemed possible that we might pick up compensatory mutations in the extracellular loops of INX-14 or INX-21. It probably would have been difficult to get a mutation in another innexin entirely—we don't know of any other innexins expressed in the germline. There is at least one other innexin expressed in the sheath (INX-10), but this innexin is likely involved in the sheath–sheath gap junctions that coordinate contractions of the myoepithelial sheath cells. Formally we have not proven that INX-10 cannot pair with INX-14/INX-21, but the phenotypes we are following—germ cell proliferation and progeny production—are completely eliminated with loss of INX-8/9, despite the presence of the other sheath innexin It might not be easy for a single mutagenic change to enable functional pairing with INX-14/21. We were hoping we might get lucky and perhaps even pick up a bypass mutation in a biochemical pathway essential to proliferation, though this was a longshot. We have added sentences in the text to help expand understanding of our motivation for carrying out the suppressor screen.
- I do not see any mention of the status of the distal tip cell in these mutants, before or after rescue? Can’t you see it in your marked strains?
Response: The distal tip cells do not appear to be affected by these suppressor mutations. The fact that the gonad arms reflex properly, as seen in Figure 2 indicates that DTC migration during gonad development is normal. The inx-8p::mCherry marker is expressed in both DTC and sheath, and expression of this marker in the DTC also appears normal. We did not add any specific comment in the text about the DTC because we have no reason to suspect that it has been affected.
- Table 1. Please explain why the germline in some rescued mutants can show so many germ cells per arm yet produce so few viable progeny. Are there also defects in sperm viability? Can the “delayed germline” phenotypes be rescued via mating with wild type males to provide better sperm?
Response: In regards to Table 1, the observation that some mutants can produce many germ cells but few progeny is at the crux of this study, thank you for bringing it up! At face value we think this suggests that the molecules going through channels required for germ cell proliferation are different (or a subset) of those required for oocyte “quality;” alternatively, it's possible that it may just be a matter of quantity, and the compromised channels formed by the suppressor mutants may allow required molecules to “dribble through” at a rate sufficient to support germ cell proliferation but not growth/viability of (relatively huge) oocytes. We favor a model for channel specificity in the mutant hemichannels because T239I, D24N tends to have smaller numbers of germ cells but higher brood sizes than T239I, E350K (no broods). This would also be consistent with speculation that functional suppression by E350K results in an increased pore size but less relief of the T239I blockage. This is all speculative, but we have added some consideration of this point in the new Discussion.
Your question about sperm, though, is one that we also had, related to the D24N mutant. In our previous paper (Starich et al., 2020), we show that gametogenesis of both sperm and oocytes is delayed in the T239I, D24N background, and once fertilization begins it continues without producing too many dead eggs (therefore there doesn't appear to be a sperm problem as the sperm can promote meiotic maturation, ovulation, and fertilization). We have added more explicit mention of this in the manuscript without specifically referring to sperm—the fact that progeny and few dead eggs are produced implies that both sperm and oocytes are functional.
- Figure 3 could be improved by indicating the cytoplasmic and apical sides of the membrane (maybe indicate intracellular vs extracellular), and the locale of the putative “dome.”
Response: Thank you for the suggestion. We have added the requested changes to Figure 3.
- In describing the phenotype of the I36N rescue strain, you mention that nuclei appear to swell at one point. Are these the sheath cell nuclei, or germline cell nuclei, or both?
Response: Regarding I36N and nuclear swelling: this is pictured in what is now Figure 7A–C. We have added a small arrow to Figure 7A that points out the nucleolus within an enlarged nucleus of one of the Sh5 cells. We don't believe there is anything specific to gap junctions indicated by the swelling of the nuclei, just that this occurs after the plasma membrane becomes leaky, and there is probably a large osmotic imbalance that affects the nucleus as well. For our purposes this swelling greatly aided our ability to recognize that the Sh5 cells were the first cells developmentally to show the I36N-induced defect (although sheath nuclei can be identified by Nomarski microscopy in wild-type animals, it is not so easy). We cannot say for certain that germ cell nuclei swell, because they are much smaller. We imagine that once the Sh5 cells start to have problems, components from these cells may leak out and adversely affect neighboring cells. We see fluid-filled foci in places outside of the Sh5 area in older animals in I36N mutants (but earlier in S9L, cf. panel E) and we cannot be sure if that is due to dead Sh5 cells spilling out toxic components, or leaky hemichannels eventually affecting other cells. (We have added to the Figure legend that the gonads pictured are from young adults.)
This also speaks to your question about sperm being limited to the gonad arm. The photo you cite (now Figure 7, panel A) shows that sperm have developed, and a –1 oocyte has grown to a decent size. In this mutant animal, sperm never gain access to spermatheca because ovulation does not occur (sperm can only enter the spermatheca upon the dilation of the distal constriction of the spermathecal, which happens during ovulation). These animals are completely sterile. We have clarified that the arrow in panel B is pointing to the Sh5 cellular membrane, showing that the cell has become completely filled with fluid.
- In the long run, it will be interesting to introduce targeted changes to the INX-9 channel in the same place as the T239I locale, to see if that would cause a similar germline phenotype in worms expressing only INX-9 in the sheath with INX-8(0).
Response: Thank you for your suggestion about attempting to recapitulate the T239I mutation in INX-9. We expect this would be the case, although we might be surprised. Along these lines, we think it would also be interesting to try to determine a residue that E350K interacts with and disrupt this amino acid by site-directed mutagenesis in a T239I background; our prediction would be a phenotype similar to T239I, E350K. Such a mutation might have shown up in our screen, but although our screen was approaching saturation it certainly did not reach saturation. We are also interested in whether there are differences in potential phosphorylation sites between INX-8 and INX-9 that may provide some specialization of function.
- Is there anything known about other membrane proteins that make up the rest of the dome?
Response: We are not aware of any more studies of the cytoplasmic dome than what has been published by Professor Oshima’s laboratory.
Reviewer 3 Report
Gap junctions in C. elegans are essential for gamete production as well as for fertilization. They allow interaction between tissue of the somatic gonad and the germ cells. Gap junctions are made of innexins. There are somatic innexins inx-8 and inx-9 and they are specific germline innexins, both forming hemichannels that couples to form a functional gap junction. Both are essential for normal reproduction of the worm. Loss of function mutation of somatic innexins leads to absence of germ cell proliferation. This severe loss-of-function phenotype precludes studies of innexin functions on the different stages of germ cell development.
To dissect the function of somatic innexins during germ cell development, a genetic approach was used to generate different forms of functional somatic hemichannel. An hypomorph allele of inx-8, carrying a missense mutation was used to screen for mutations that restore fertility of the worm. Six suppressor mutations were isolated. All the suppressor mutations are rescuing proliferation and fertility to different extent and the authors provide interpretations based on hemichannel 3D structure established previously on how these different intragenic suppressors could affect gap junction function.
This study is a very useful contribution to the field of gap junctions and provides new tools to dissect further the role of innexins in the production of functional gametes.
Minors comments:
-Material and methods line 147 or in the text line 355: The reference (ref 11) to the inability of inx-8(D24N) to form gap junction is missing
-In Table 1: Indicating the brood size of the starting strain inx-8(tn1513 T239I) would help to get the suppression effect at glance
Author Response
Response to Reviewer 3
Thank you for taking the time to read our manuscript and provide comments.
- The reference to the inability of inx-8(D24N) to form gap junction is missing.
Response: This comment refers to the inability of D24N mutation in INX-8 to form junctions. That information is first reported in this manuscript. We put a more thorough explanation of that analysis in the Materials and Methods for those who might be interested, and a more cursory statement in the Results. We did not feel that this information merited an additional figure, especially since it is a negative result (we thought it would not be so helpful to show an image of an embryo lacking GFP expression).
- In Table 1: Indicating the brood size of the starting strain inx-8(tn1513 T239I) would help to get the suppression effect at glance
Response: This was a good suggestion, thank you. We have added the brood size data of T239I to Table 1 as suggested.
Round 2
Reviewer 1 Report
I have no further comments.